# Sex Hormones and Metabolic Dysfunction-Associated Steatotic Liver Disease

**DOI:** 10.3390/ijms26199594

**Published:** 2025-10-01

**Authors:** Ralf Weiskirchen, Amedeo Lonardo

**Affiliations:** 1Institute of Molecular Pathobiochemistry, Experimental Gene Therapy and Clinical Chemistry (IFMPEGKC), RWTH University Hospital Aachen, D-52074 Aachen, Germany; 2AOU Modena, Department of Internal Medicine, Ospedale Civile di Baggiovara (-2023), 41100 Modena, Italy

**Keywords:** fibrosis, cirrhosis, HCC, MALO, sex differences, steatosis, steatohepatitis

## Abstract

Positioned at the intersection of sex medicine and endocrinology, metabolic dysfunction-associated steatotic liver disease (MASLD) is often managed by specialists who may not be fully familiar with the complex roles of sex hormones in its pathogenesis and clinical course. To address this gap, we review the molecular actions of testosterone, estradiol, and progesterone on liver functions, as well as the role of sex-hormone binding globulin (SHBG) in MASLD histogenesis, highlighting disparities by sex as well as reproductive status. We also discuss how sex hormones influence fatty acid metabolism, gut dysbiosis, mitochondrial activity, gluco-lipidic homeostasis, lipotoxicity, inflammation, and MASLD-related liver tumorigenesis. Furthermore, we examine observational studies on associations between endogenous and exogenous sex hormones and SHBG with MASLD, with attention to hypogonadism in either sex or polycystic ovary syndrome. We summarize the role of sex hormones in modulating MASLD risk across life stages such as menopause, breastfeeding, and lactation. Lastly, we review the hepatic effects of hormone replacement therapy (HRT) on MASLD in both sexes, prospects, and safety of HRT and contraceptives among individuals with chronic liver disease. In conclusion, sex hormones play significant roles in MASLD pathobiology, underscoring the importance of sex-specific approaches in clinical practice and research.

## 1. Introduction

Due to declining rates of chronic viral hepatitis and the rise in obesity and diabetes, metabolic dysfunction-associated steatotic liver disease (MASLD) is now the leading cause of chronic liver disease globally [1,2]. In obesity, expanded, inflamed, and dysfunctional adipose tissue acts as an endocrine organ contributing to MASLD through a deranged adipokine profile [3]. Type 2 diabetes (T2D), the prototypic endocrine disorder associated with MASLD, is strongly linked to MASLD initiation and progression, predisposing to the development of T2D [4]. MASLD is the latest nomenclature used to describe steatotic liver disease (SLD) due to perturbed metabolic balance and its bidirectional nexus with the metabolic syndrome [5]. Its positive identification (“metabolic dysfunction-associated”) emphasizes metabolic dysfunction, which is under close hormonal control at the systemic and cellular level [6]. Various hormones regulate the multi-step biochemical cascades that promote de novo lipogenesis (DNL) [7,8]. These steps eventually lead to the buildup of liver fat, with a relatively impaired beta-oxidation of fatty acids and a proportionally reduced capacity to export triglycerides from the hepatocyte [9,10,11].

Both DNL and increased hepatic influx of free fatty acids (FFA) due to peripheral insulin resistance contribute to steatosis. Furthermore, in the case of the latter, synthesizing triglycerides is safer than the accumulation of FFAs or diacylglycerols, both of which are toxic to hepatocytes.

In MASLD, systemic metabolic dysfunction is influenced by growth hormone (GH), thyroid hormones (THs), and other endocrine axes [8]. Epidemiological studies unequivocally indicate that sex hormones and reproductive status are strongly linked to the initiation and advancement of MASLD through their impact on body fat distribution, systemic, and organ-specific metabolic dysfunction [12] and by modulating immune responses to sterile inflammation and hepatocyte injury. During the reproductive age, MASLD is more prevalent and severe in men than in women. However, after menopause, MASLD prevalence increases in women, who also have a greater risk of advanced fibrosis than men, suggesting a protective effect of estrogen [13]. Sex differences also exist in the major risk factors of MASLD, as well as in disease prevalence, fibrosis, and clinical outcomes [14]. Moreover, hormonal factors, by interacting with inherited gene variants, impact MASLD development and progression [15].

Together with gender (i.e., sociocultural attributes), sex is a key modifier of health, disease, and medicine [16]. Therefore, the sexual dimorphism observed in MASLD is also seen in many other human diseases, such as cardiovascular diseases, which often show significant differences between men and women in pathophysiology, epidemiology, and clinical outcomes [16]. Efforts by health policymakers have led to an increasing number of publications addressing sex differences in various fields related to MASLD, like T2D, coronary artery disease, and immunology [17]. The systemic nature of MASLD, its pathogenic heterogeneity, and the range of liver-related and extra-hepatic manifestations strongly support the need for a sex-specific approach in MASLD and metabolic dysfunction-associated steatohepatitis (MASH) trials, which is an important yet unmet research need [18,19,20].

Overall, these considerations highlight that MASLD sits at the intersection of sex medicine and endocrinology, suggesting that sex hormones play a crucial role in sex-specific therapy and precision medicine strategies. Therefore, this review aims to explore the insufficiently defined impact of sexual dimorphism in MASLD itself, as well as in its hepatic and extra-hepatic manifestations and complications. Incorporating insights into hepatic sexual dimorphism into medical practice holds promise for refining diagnostics, guiding the development of more individualized and effective therapeutic strategies, and ultimately improving outcomes in patients with MASLD [21]. While acknowledging that sexual chromosomes may also influence the pathogenesis of metabolic disorders and MASLD [15], this review will focus on analyzing the epidemiological features, pathogenic determinants, and therapeutic implications of sex hormones and reproductive status in MASLD.

### Method of Bibliographic Research

A comprehensive literature review was conducted in PubMed/MEDLINE, Embase, Web of Science, and Scopus, covering the period from January 1990 to July 2025. Search strings combined controlled vocabulary (MeSH/Emtree) and free-text terms for “MASLD OR MAFLD OR NAFLD”, “sex hormones”, “testosterone”, “estradiol”, “progesterone”, “sex-hormone-binding globulin”, “growth hormone”, “thyroid hormone”, “sexual dimorphism”, “pregnancy”, “lactation”, “menopause”, “hypogonadism”, and “hormone replacement therapy”. Reference lists of retrieved articles were manually screened to include additional publications. Basic science reports, animal experiments, observational studies, randomized trials, Mendelian-randomization analyses, and systematic reviews written in English were included. Data were extracted into pre-specified tables and cross-checked for accuracy. When overlapping cohorts were identified, the most comprehensive or recent publication was retained. The term MASLD was used in accordance with the latest consensus definition, emphasizing metabolic dysfunction as the primary driver of steatosis.

## 2. Molecular Physiopathology of Sex Hormones and Their Receptors in the Context of MASLD

### 2.1. The Liver as a Sexually Dimorphic Organ

The liver is a crucial organ in maintaining energy, protein, glucose, lipid, and cholesterol balance. It also plays a role in controlling blood volume, supporting the immune system, regulating endocrine control of growth signaling pathways, and metabolizing xenobiotics [22]. Moreover, significantly influencing hepatic metabolic zonation, sexual dimorphism of the liver has the potential to explain observed differences between sexes in susceptibility, and progression of MASLD and MASLD-related hepatic and extra-hepatic events [21,23]. While advanced technologies, such as single-cell RNA sequencing and digital twin modeling, have enhanced our understanding of the molecular basis of hepatic sexual dimorphism, the pathomechanisms driving sex differences in MASLD remain incompletely defined [21,24].

Sex hormones and sex chromosomes contribute to physiological differences and disease variation between men and women. Hormonal pathways, including those involving estrogen and androgen, affect the development of liver disease. Additionally, genetic differences on sex chromosomes may underlie variations in metabolism, gene expression, and immune response [24].

A multistep signaling cascade is initiated in the arcuate nucleus, which produces growth hormone-releasing hormone (GHRH) to stimulate GH release, and the ventromedial preoptic nucleus, which is part of the preoptic area/anterior hypothalamus that regulates GHRH and somatostatin expression and release. This cascade eventually regulates the dynamic patterns of GH secretion in a sex-specific manner [25]. This hormonal control is then transmitted to the liver, where GH receptors are located [25,26]. The timing and pattern of pituitary GH secretion, episodic in males and more stable in females in many species, play a key role in establishing and maintaining differences between sexes in hepatic gene transcription [26]. The liver exhibits substantial sex-specific variations in the expression of more than 1000 genes related to the metabolism of steroids, lipids, and xenobiotics [26]. Approximately 90% of these sex-specific liver genes depend largely on sexually distinct patterns of GH secretion, suggesting a mechanism in which the central nervous system indirectly regulates sex-related differences in liver activity associated with steroid metabolism through GHRH and somatostatin secretion [26]. GH phosphorylates, activates, and leads to the DNA binding of signal transducer and activator of transcription 5b (STAT5b), which also has the same pulsatile activity pattern in males [27,28]. Additionally, transcription factors such as hepatocyte nuclear factor 4 alpha (HNF4a), B-cell CLL/lymphoma 6 (BCL6), and zinc fingers and homeoboxes 2 (ZHX2) also modulate GH-STAT5 signaling [29,30,31]. Together with the sex-dependent expression of GH, testosterone, and estradiol signaling pathways are the primary regulators of hepatic sexual dimorphism [32]. Of note, sex differences in GH secretion become less pronounced after age 50 [33], suggesting that hepatic gene expression profiles may change significantly in women after menopause. This underscores the importance of accounting for both sex and menopausal status when designing hepatic transcriptome analyses.

The functional interaction between GH and steroid hormone regulation has allowed for the modeling of female and male human hepatocytes [34]. Finally, a novel pathway of estrogen-mediated crosstalk between hepatocytes and hepatic stellate cells (HSCs) may contribute to sex disparities in MASLD via an anti-fibrogenic activity of the S1P/S1PR3 axis [35].

### 2.2. Molecular Mechanisms of Sex-Specific Action of Testosterone on the Liver

Sexual dimorphism in hepatic physiology and disease is increasingly attributed to differential androgen signaling. Testosterone, acting through the hepatic androgen receptor and intersecting metabolic pathways, influences lipid handling, immune activation, oxidative stress, and fibrogenic remodeling in MASLD/MASH. In this section, we will outline the molecular basis of these sex-specific actions, with a focus on lipid droplet biology and susceptibility to ferroptosis, and emphasize the protective effects of physiological testosterone in males. Physiological doses of testosterone affect all three individual components of MASH histology, protecting against steatogenesis, blunting hepatic inflammation, and exerting anti-fibrotic activity (Figure 1).

#### 2.2.1. Lipid Droplets

LDs are spherical, cytoplasmic organelles with a central core of neutral lipids surrounded by an external monolayer of various endoplasmic reticulum membrane-derived phospholipids and proteins that stabilize the molecular structure of LDs, participate in the processing of accumulated lipids, and play key roles in their pathobiology [36]. By dynamically interacting with mitochondria and efficiently delivering fatty acids, LDs serve as master regulators of lipid metabolism and energy production under physiological conditions [36]. However, their disrupted homeostasis due to unmatched synthesis overcoming LD degradation induces the storage of large LDs in hepatocytes, which is critical in the pathogenesis of SLD [37]. Intrahepatic LD accumulation is an adaptive response to the liver being overloaded by an excess of lipid and carbohydrate substrates from the bloodstream. It prevents lipotoxicity and oxidative stress from excess free fatty acids, protects against lipid peroxidation of cell and organelle membranes, and facilitates the disposal of damaged proteins and lipids [37].

A comparative study of β-estradiol and testosterone in palmitate/oleate-induced fatty accumulation in male-derived HepG2 cells has found that both hormones significantly influence LD accumulation and regulatory gene expression, with β-estradiol exhibiting a stronger effect on promoting lipid turnover and reducing lipid buildup [38].

#### 2.2.2. Ferroptosis

Hepatocyte death is crucial in the pathogenesis of MASLD through mechanisms that are not fully understood. Ferroptosis, an iron-dependent regulated cell death process, is harmful in MASLD patients. Non-PUFA supplementation to hepatocytes, which alters the composition of the lipid bilayer, contributes to maintaining sensitivity to ferroptosis in male mice [39]. Interestingly, testosterone deficiency has been shown to affect ferroptosis and circadian rhythm-related signaling pathways in a genetically engineered male mouse model [40], suggesting that testosterone deficiency may influence MASLD pathogenesis through ferroptosis. However, further studies are needed.

### 2.3. Molecular Mechanisms of Estradiol Action on the Liver

The extensive literature on the role of estrogens in MASLD pathogenesis demonstrates that female sex hormones interact at various stages throughout the progression of MASLD. This includes development, inflammatory-fibrotic progression, immune modulation, and the formation of MASLD-related tumors. In vivo, hormonal effects may differ depending on reproductive status, particularly baseline estrogen levels. Therefore, it is crucial to critically analyze the variable effects before applying findings from animal studies to humans, taking into account sex- and menopause-specific influences.

#### 2.3.1. Estradiol Interacts with Genetic Risk Variants

Sex hormones interact with genetic risk variants to modulate the initiation and trajectory of MASLD. Two prominent examples are *PNPLA3* I148M and HSD 17B13. *PNPLA3* I148M is a “gain of function” variant that enhances the accumulation of lipids within hepatocytes and carries an increased risk of MASLD development [41]. It interacts with estrogen signaling through estrogen receptors alpha (ERα), resulting in sex-specific effects of PNPLA3 variants on MASLD [42]. Cherubini et al. [43] conducted a study with a complex experimental design that collected data from menopausal women, obese individuals, and experimental mice. This investigation found that the p.I148M variant was associated with a higher risk of MASLD development and progression among women compared to men. In obese subjects, transcriptomic analysis indicated that insulin resistance, the p.I148M variant, and female sex were strongly associated with higher expression of *PNPLA3*. Women carrying the variant showed increased expression of genes regulating inflammation and fibrosis pathways. In a mouse model, the hepatic expression of *PNPLA3* was higher in females during the estrogenic phase rather than during the luteal phase or in males. This finding is compatible with the notion that estrogens may modulate *PNPLA3* expression in the liver.

HSD 17B13 is a “loss of function” variant implicated in the biogenesis of lipid droplets and metabolism of retinoid and sex hormones, protecting from MASLD progression. This beneficial effect is more evident in postmenopausal women, particularly when estrogen levels physiologically decline, as premenopausal women are already protected by elevated estrogen levels [44].

#### 2.3.2. Effects on Gluco-Lipidic Homeostasis, Lipotoxicity, Inflammation, Mitochondrial Function, and Oxidative Stress

Estradiol signaling protects Female ApoE KO Mice against Western-Diet-Induced MASH and glucose intolerance [45]. By improving mitochondrial function, this hormone mitigates fatty acid-induced insulin resistance in hepatocytes by downregulating JNK activation. In contrast, PPARα, which promotes fatty acid oxidation [30], was increased only in female rats in response to HFD feeding [46]. Consistently, hepatic PGC1s are important regulators of intrahepatic ROS detoxification, and female rats are particularly sensitive to their loss due to essential roles in estrogen/ERα signaling. The loss of estrogen signaling contributes to oxidative damage caused by low levels of PPARG coactivator 1α (PGC1A) in the liver, worsening steatohepatitis induced by diets rich in fat and fructose [47].

The estrogen-related receptor (ERR) family of orphan nuclear receptors contains transcriptional activators for genes involved in the biogenesis and energetic homeostasis of mitochondria [48]. Estrogen-related receptor alpha (ERRα) regulates key metabolic processes, such as hepatic VLDL-TG assembly and output into the bloodstream, through target genes (*Apob*, *Mttp*, *Pla2g12b*) in a sex-specific manner, thereby facilitating the capacity of the hepatocyte to export lipids into the bloodstream. In liver-specific ERRα-deficient mice, reduced hepatic VLDL-TG secretion leads to SLD, increased ER stress, and inflammation, contributing to MASH development [49]. ERRα functions downstream of estrogen/ERα signaling to influence sex-dependent variations in hepatic lipid homeostasis. In agreement, a recently developed small-molecule inhibitor for ERRα can block MASLD development due to high-carbohydrate or high-fat diets, reduce lipid accumulation, and attenuate MASH development in *Pten* null mice [48]. Finally, glycerophosphate acyltransferase 4 was identified as a novel transcriptional target of ERRα, indicating that glycerolipid synthesis is another mechanism involved in the initiation of MASLD/MASH regulated by ERRα [48].

#### 2.3.3. Effects on the Metabolism of Fatty Acids, Fibrogenesis, and Gut Dysbiosis

Surgical or natural menopause predisposes individuals to MASLD, a risk that can be reduced by hormone replacement therapy (HRT). Various physiological variables and disease progression influence the gut microbiota. This microbial community lives in symbiosis with the host, modulating immunity, contributing to endocrine-metabolic homeostasis, and enforcing defense mechanisms. Supplementation of estradiol prevents MASLD in ovariectomized rats (OVX) [50]. OVX-induced alterations in fatty acid metabolism are partially restored by E2 supplementation through changes in gut microbiota. Linoleic acid (LA) metabolism is disrupted in MASLD, with gut microbiota associated with abnormal biomarkers and different LA metabolites closely linked to MASLD [50]. LA induces SLD in vitro, and E2 administration normalizes the transcriptional changes in the LA metabolism pathway [50]. These findings suggest that post-menopausal MASLD is directly and indirectly affected by the dysfunctional metabolism of LA.

OVX-Spontaneously Diabetic Torii (SDT) fatty rats recapitulate SLD pathobiology and exhibit more severe fibrosing MASH than control animals, suggesting that estrogen reduction caused by OVX promotes induced fibrosing MASH through increased de novo intrahepatic lipogenesis or, hypothetically, impaired VLDL secretion [51]. ERα plays a key role in mediating sex-specific responses to high-fat diets and is associated with differing effects on hepatic health in mice of both sexes [52].

Adropin, a hepatokine regulated in a sex-specific and ERα-dependent manner, increases in female mouse livers under high-fat diet conditions. This hepatic induction of adropin, driven by ERα, is inversely correlated with lipogenic gene expression and SLD, suggesting that ERα-mediated adropin may help prevent MASLD in females [53].

Estrogen reduction, similar to that seen in menopause, is associated with gut dysbiosis and alterations in intestinal microbiota metabolites such as short-chain fatty acids (SCFA), along with a disorder in antimicrobial peptides production and lipid metabolism, promoting MASLD [54].

#### 2.3.4. Masculinization of the Liver in Post-Menopausal Women

To clarify the role of impaired ERα signaling in post-menopausal MASLD, Meda et al. [55] examined the liver transcriptomes of sham-operated and OVX control and liver ERα knockout (LERKO) female mice by performing RNA-Seq analysis. The data revealed that ovariectomy induced male-biased genes, especially in the liver of control females, indicating that hepatic ERα is indeed involved in the masculinization of the liver that occurs after estrogen loss. Furthermore, obese women with MASLD, particularly those aged 51 years and older, also exhibited a male-like shifted profile, suggesting that the masculinization of the female liver contributes to the development of MASLD in women with obesity. The study identifies hepatic ERα as a promising target for the prevention and management of post-menopausal MASLD.

### 2.4. Role of Sex Hormones in the Biogenesis of MASLD-Related Liver Tumors

Sex hormones differentially influence all hepatic cell lineages, including hepatocytes, endothelial cells, HSCs, and Kupffer cells (KCs) [56]. In hepatocytes, estrogens promote DNA synthesis and upregulate endothelial nitric oxide synthase in endothelial cells. They also inhibit the release of pro-inflammatory cytokines and oxygen radicals from KCs, and reduce endothelin levels, thereby decreasing HSC contraction. Androgens increase hepatocyte mitogenesis, stimulate angiogenesis by acting on sinusoidal endothelial cells, stimulate collagen deposition, and are pro-inflammatory [56]. Estrogens, particularly 17β-estradiol (also known as E2), the most active form of the estrogen family, are major contributors to protecting liver health (Figure 2).

ERα downregulates inflammation and inhibits IL-6 release from KCs, which is involved in the male prevalence of hepatocellular carcinoma (HCC), through the modulation of NF-κB. This contributes to hepatic carcinogenesis by activating signal transducer and activator of transcription-3 signaling [56,57]. Additionally, some microRNAs expressed by HCC cells downregulate ER, leading to hormonal unresponsiveness in cancer cells [58]. Despite this evidence, in certain conditions, estrogens may potentially favor HCC development through the formation of free radical-mediated DNA and RNA adducts, which have a mutagenic potential. Endogenous and exogenous androgens favor liver cancer development through Ars [59]. Finally, sex hormones are also involved in the development and outcomes of cholangiocarcinoma (CCA), especially intrahepatic CCA, as reviewed elsewhere [56,60,61]. The dual roles of sex hormones in tumor initiation and progression emphasize that their hepatic actions cannot be viewed in isolation. Protective anti-inflammatory pathways (e.g., ERα-mediated NF-κB repression) coexist with pro-carcinogenic mechanisms driven by oxidative DNA damage, AR signaling, or microRNA-induced ER loss. Similar bidirectional themes appear in other liver pathologies, such as CCA.

Therefore, the effects of testosterone and estrogen on liver health and disease are multifaceted and context dependent. They influence metabolic pathways, inflammatory responses, and fibrosis in both protective and potentially deleterious ways. For an integrated overview of these diverse actions in the pathogenesis of testosterone and estrogen-related fat-associated diseases that can be causative for the formation of hepatic tumors, please refer to Figure 3 and Figure 4.

### 2.5. Molecular Mechanisms of Action of Progesterone on the Liver

Progesterone exerts its hepatic effects mainly via nuclear progesterone receptors (PR-A and PR-B) and membrane-bound progesterone receptors, which modulate gene transcription and rapid non-genomic signaling cascades. In hepatocytes, PR-B activation enhances the activity of key lipogenic enzymes, including acetyl-CoA carboxylase and fatty-acid synthase, stimulating DNL and favoring triglyceride deposition within LDs [63]. Recent in vitro and murine studies have shown that supraphysiologic levels of progesterone raise hepatic SREBP-1c activity, suppress PPARα signaling, and down-regulate genes involved in β-oxidation, creating a pro-steatogenic environment [63]. Research on isolated rat liver mitochondria has demonstrated that progesterone and its synthetic analogues can impact various physiological functions of mitochondria and the production of reactive oxygen species (ROS), inhibiting NAD-dependent respiration, and affecting the mitochondrial respiratory chain [64]. Furthermore, progesterone reduces the calcium retention capacity of mitochondria, leading to the opening of the mitochondrial permeability transition pore (MPTP), a crucial regulator of cell death [65]. Progesterone can also worsen insulin resistance by inducing apoptosis in pancreatic β-cells through oxidative stress mechanisms [64,65] and inhibiting the PI3-kinase pathway by multiple steps that include the expression of IRS-1 and distal PI3K–AKT signaling pathway, ultimately prompting hepatic fat accumulation [66]. Conversely, in other organs, the expression of PRA and PRB significantly correlates with the expression of NRF2, which plays a crucial role in cellular defenses against metabolic, xenobiotic, and oxidative stress by reducing mitochondrial ROS [67]. The dual actions of progesterone, being steatogenic at high concentrations yet potentially cytoprotective at physiological levels, may explain the variability in clinical findings among women exposed to different progestin therapies, highlighting the need for dose- and context-specific evaluation in MASLD.

Studies in rats have shown that ovariectomy leads to lipid peroxidation in liver tissues, with additional pinealectomy further increasing hepatic lipid peroxidation. However, supplementation with estradiol and progesterone significantly protects against lipid peroxidation, suggesting a protective role of progesterone in liver tissue [68]. Additionally, progesterone administration can increase lipid synthesis rates in isolated fetal hepatocytes [69]. On the other hand, the loss of compounds relevant for progesterone receptor function promotes SLD through the induction of DNL in mice, indicating that the impact of progesterone on SLD is complex and not fully understood [70].

### 2.6. Role of GH in Explaining Sex Differences in MASLD

The sexual dimorphism in the pattern of GH secretion is orchestrated by hypothalamic GHRH and somatostatin neurons [71,72]. The liver is the primary target of circulating GH, where binding of the ligand activates the JAK-STAT5 pathway. In male mice livers, the typical physiological pulses of endogenous GH stimulate repeated rounds of chromatin opening and closing, directing the transcriptional activation of GH-activated STAT5 through the activation of sex-biased genes. Conversely, in female livers, STAT5 is consistently activated by the near-continuous presence of circulating GH, resulting in a significant enrichment of STAT5 binding near female genes [73,74].

In males, intermittent STAT5 activation promotes IGF-1 synthesis, VLDL-triglyceride export, and maintains lower hepatic fat [75]. On the other hand, the continuous activation in females supports higher activity of genes regulating fatty acid uptake and storage [76]. Experimental models of GH deficiency or resistance consistently develop steatosis, inflammation, and fibrosis that mimic MASH, with more severe liver injuries in male rodents [77]. This finding is in line with evidence showing that increased GH levels achieved either by raising endogenous GH secretion or administering exogenous GH replacement therapy enhance IGF-1 production and reduce steatosis and the severity of liver injury in MASLD [78,79,80]. Clinically, adults with isolated GH deficiency or acromegaly treated with somatostatin analogs display increased MASLD prevalence [81]. However, more recent studies have shown that susceptibility to acromegaly itself depends on genetic factors [82].

Interestingly, GH replacement therapy reverses steatosis and improves insulin sensitivity [11,83,84]. These observations highlight GH as a key modulator of hepatic sexual dimorphism and suggest that disrupted GH signaling may amplify, or even override, the protective effects of estrogens in pre-menopausal women. Potential targets in this context could be PGC1A, PPARα, FXR, and LXR, which act as master regulators involved in the initial step of steatogenesis, the accumulation of triglycerides [34].

### 2.7. Potential Interaction of Sex Hormones and Thyroid Hormones

Thyroid hormones (THs) and sex steroids intersect at various metabolic steps in the pathogenesis of MASLD. THs increase basal metabolic rate, promote mitochondrial β-oxidation and boost LDL-receptor expression, while hypothyroidism can lead to steatosis and dyslipidemia [85]. Estrogens enhance hepatic deiodinase-1 expression, increasing local triiodothyronine (T3) levels [86], while androgens seem to decrease TH transport proteins, subtly changing TH distribution [87]. Conversely, THs impact sex-steroid production by affecting gonadotropin-releasing hormone secretion and directly regulating steroidogenic-enzyme expression in the gonads and adrenals [88]. T3 can modulate liver metabolism by activating peroxisome proliferator-activated receptor-γ and liver X receptor [89]. In hepatocytes, T3 and estradiol work together to activate PGC-1α and AMP-activated protein kinase, supporting fatty-acid oxidation and autophagy [90]. This hormonal cooperation diminishes after menopause, contributing to the increased risk of MASLD in post-menopausal women [90]. Additionally, lower TH levels can intensify the steatogenic effects of hyperandrogenism in women with polycystic ovary syndrome (PCOS) [91], demonstrating a complex interplay of hormones that influences individual susceptibility to MASLD. Understanding these interactions could lead to new treatment approaches combining sex-hormone modulation with selective TH-receptor-β agonists to enhance hepatic fat and fibrosis reduction.

## 3. Lesson from Epidemiological and Meta-Analytical Studies

### 3.1. Testosterone and Estrogens in Men and Women

Recent epidemiological, observational, and meta-analytic reviews summarizing the role of serum concentrations of endogenous sex hormones and sex hormone-binding globulin (SHBG) in MASLD development and progression are listed in Table 1 [92,93,94,95,96,97,98,99,100,101,102,103,104,105,106,107,108,109,110,111]. Most published studies are either cross-sectional or retrospective, indicating the need for longitudinal and prospective studies. Moreover, the identification of MASLD varies in diagnostic techniques used, including non-invasive biomarkers, conventional ultrasonography, non-contrast computed tomography, fibroscan, and liver biopsy. Additionally, there is significant heterogeneity in the number of participants recruited, ranging from children and adults to individuals from the general population, those in the histological USA database, those with obesity undergoing bariatric surgery, and individuals with T2D. Collectively, these methodological differences make it challenging to draw any meaningful comparisons among the studies, leading to variable and sometimes conflicting findings, highlighting the need for further investigation. Despite these limitations, it can be reasonably concluded that endogenous sex hormones and SHBG are linked to MAFLD in both adults [102] and boys with obesity and MAFLD [111].

More specifically, in men:Serum total testosterone (TT) is strongly associated with MASLD in men with T2D [104].Lower TT is associated with MASLD among individuals with T2D and more severe MASLD/MASH [98,101,103].Consistently, higher serum TT is negatively associated with a lower prevalence of MASLD among men with T2D [105].

In women:Estrogens protect from liver fibrosis pre-menopausally, and this protection is lost with menopause [92,93]. The risk of liver fibrosis is directly proportional to the duration of estrogen deficiency [93].Ovarian reserve assessed with Anti-Müllerian hormone (AMH) is associated with the histological severity of MASLD [109].SHBG protects from liver fat accumulation [108].Lower SHBG and higher bioavailable testosterone levels confer an increased MASLD risk [94,110].

However, the use of exogenous sex hormones is associated with an increased risk of MASLD and variable liver histology outcomes in pre- vs. post-menopausal women [95,99].

PCOS, defined by clinical and/or biochemical evidence of hyperandrogenism, chronic oligo-anovulation, and polycystic ovary at ultrasonographic scanning, provides further evidence for the biological role of sex hormones in MASLD [112]. In women with PCOS, hyperandrogenemia can adversely affect systemic and hepatic metabolism, elevating the risk of dysmetabolic states and MASLD [113]. Insulin resistance and visceral obesity, common among women with PCOS, further strain liver health [114].

Finally, TT, SHBG, and free testosterone concentrations are implicated in the risk of primary liver cancer. A study including data from approximately 200,000 males and 180,000 postmenopausal females participating in the UK Biobank and followed for a median of 11.8 years has shown that TT and SHBG were positively associated with liver cancer risk in either sex, whereas free testosterone was inversely associated with primary liver cancer in males and not associated with the risk of liver cancer among females. Further analysis comparing HCC and intrahepatic cholangiocarcinoma (ICC) disclosed that TT and SHBG concentrations were only positively associated with HCC (but not with ICC) in both sexes [115]. These findings indicate the need for further research into the mechanisms through which sex steroids influence the risk of primary liver cancer. 

### 3.2. Sex Hormones, Liver Enzymes, and Cardiometabolic Factors

A close and bidirectional relationship exists between T2D and MASLD. The regulation of postprandial glucose (PPG) is closely linked to insulin secretion, insulin resistance, and beta-cell function, all of which are intimately implicated in the development of T2D [4,116]. Post-meal glucose responses to an oral glucose tolerance test can vary among individuals due to factors such as body fat, lipid profile, liver enzymes, and steroid hormones, with observed differences between men and women. Masango et al. identified the sets of sex hormones, hepatic enzyme levels, and cardiometabolic factors associated with PPG in individuals of African ancestry [117]. They found that while polygenic risk did not impact PPG variability, a cluster of sex hormones, liver enzymes, and cardiometabolic factors explained approximately 10% of the variability in PPG in both sexes [118]. The novel approach followed in this study has contributed to precision medicine by showing that inter-individual differences in PPG responses to an oral glucose tolerance test may be accounted for in either sex by body fat distribution, lipid profiles, hepatic enzymes, and steroid hormones. Additional undefined variables may contribute to inter-person variability among populations of African ancestry. Given that these populations tend to be less prone to the risks of T2D and MASLD [118], additional studies are necessary among other ethnicities.

The line of research summarized above is potentially relevant to explaining the increasingly recognized disparities in cardiovascular outcomes owing to sex and reproductive status among those with MASLD [119,120].

## 4. Evidence from Mendelian Randomization Studies

Although epidemiological studies provide a strong foundation for defining exposure-disease associations, they have limited utility in inferring cause-and-effect relationships. Mendelian randomization (MR) uses gene variants as a natural experiment to proxy an exposure, overcoming limitations of epidemiological studies such as their observational nature, susceptibility to biases resulting from undetected confounders, and reverse causation [121]. Three recent investigations have assessed the causal role of sex hormones and SHBG on MASLD in both sexes. These studies, although limited in number, offer findings that partially align with the numerous observational studies listed in Table 1.

Cai X et al. [108] selected gene variants associated with sex hormones and SHBG and analyzed their associations with magnetic resonance imaging (MRI)-assessed liver fat content (LFC) from the most recent European genome-wide association studies (GWAS). This study revealed that higher genetically determined SHBG was linked to lower LFC in women (β = −0.36, 95% CI: −0.61, −0.12), suggesting a protective role of SHBG against liver fat accumulation specifically in women. However, no causal association was found between genetically determined sex hormones and LFC.

Liu et al. [122] conducted a bidirectional two-sample MR analysis to evaluate the causal relationship between MASLD and PCOS. They utilized data from large-scale biopsy-confirmed MASLD GWAS with 1483 cases and 17,781 controls, as well as PCOS GWAS with 10,074 cases and 103,164 controls in European ancestries. The MR mediation analysis also included data from GWAS on glycemic-related traits (up to 200,622 individuals) and sex hormones (189,473 women) from the UK Biobank to investigate potential mediating roles of these molecules in the causal connection between MASLD and PCOS. The findings indicated that genetically predicted MASLD was linked to an increased risk of developing PCOS (Odds Ratio per one-unit log odds increase in MASLD: 1.10, 95% CI: 1.02–1.18; *p* = 0.013), with mediation effects observed through fasting insulin and sex hormones. However, the study found limited evidence supporting the reverse association of PCOS causing MASLD.

Finally, Weng et al. [110] conducted a study of 187,395 men and 170,193 women from the UK Biobank followed for 12.49 years. The data showed that lower levels of SHBG and higher concentrations of bioavailable testosterone were causally associated with an increased genetic risk of MASLD in women (OR (95% CI): 0.57 (0.38, 0.87) and 2.21 (1.41, 3.26). In men, SHBG acts nonlinearly. The bidirectional MR analysis also showed the effect of MASLD on SHBG and bioavailable testosterone levels in both sexes. Therefore, MR analyses confirm the causality of the associations found in women regarding SHBG and testosterone. Additional studies should explore these associations in men.

## 5. Sex Hormones and MASLD in Pregnancy, Lactation, and Menopause

### 5.1. Pregnancy

An experimental study in mice [123] showed that female sex hormones have deleterious effects on hepatic mitochondria, suggesting that these harmful hormonal actions may contribute, together with other factors, to acute fatty liver of pregnancy (AFLP). AFLP is a rare, acute complication of pregnancy featuring microvesicular steatosis caused by mitochondrial dysfunction, making it distinct from true MASLD. These groundbreaking observations [123] provide a comprehensive understanding of the impact of female sex hormones on liver health during pregnancy.

The rates of MASLD in pregnant women are increasing, and MASLD raises the risks of pregnancy complications, including gestational diabetes, pre-eclampsia, and preterm birth [124]. Timely medical counseling should optimize metabolic balance before and during pregnancy. Moreover, fibrosis should be assessed before pregnancy. This approach will help identify cases of silent cirrhosis, conduct preconception variceal screening, and collaborate with maternal-fetal medicine specialists [124]. Additionally, to diminish the risk of preeclampsia, aspirin prophylaxis is recommended at 12 weeks of gestation in MASLD patients [124].

### 5.2. Lactation

The estrogen-responsive pituitary hormone prolactin exerts an anti-steatotic activity, as proven by the finding that genetically engineered mice knock-out for the prolactin receptor exhibit increased hepatic triglyceride accumulation [125]. Lactation is hormonally regulated, with prolactin, estrogen, progesterone, and oxytocin coordinating mammary gland development, milk production, and ejection [126]. Breastfeeding protects the mother-infant dyad from MASLD through a variety of biological mechanisms. These include, for the infant, beneficial metabolic imprinting and reduced obesity risk in adulthood, shaping the infant gut microbiota consistently with the mother’s, assimilation of components of maternal milk, including docosahexaenoic acid, which has the potential to act as anti-fibrogenic PPAR-agonist agents, human milk-derived extracellular vesicles rich in metabolically active proteins and miRNAs, which modulate insulin resistance, steatogenesis and fibrogenesis [127]. In turn, lactating mothers are protected from MASLD and metabolic syndrome because milk production diverts circulating nutrients and reduces insulin resistance [127]. Moreover, lactation utilizes pregnancy energy reserves, estimated at 321 to 325 MJ for a 12 kg weight gain, and helps restore normal metabolism after pregnancy [127].

Therefore, breastfeeding should be encouraged in the context of MASLD to prevent cross-generational disease propagation [124,127]. However, the safety of drug therapies for MASLD during pregnancy and lactation remains to be determined [124].

### 5.3. Menopause

Evidence supporting the protective role of endogenous estrogens against MASLD is found in the finding that menopause is a risk factor for its development. Strong evidence for this concept was published by Jaroenlapnopparat and colleagues [128]. Through a meta-analysis of 12 studies, these authors demonstrated a strong association between menopause and MASLD (pooled OR of 2.37, 95% CI, 1.99–2.82; I2 = 73%). This association remained significant across different age groups and metabolic confounders in a sensitivity analysis of six studies, with a pooled OR of 2.19 (95% CI, 1.73–2.78; I2 = 74%). No publication bias was observed in the funnel plot. In contrast, a previous study [129] found no significant differences in MASLD risk based on the age of menopause (normal, early, or late), but did note an increase in advanced fibrosis with longer durations of endogenous estrogen deficiency. Another study by Wegermann et al. [130] found 10 loci with significant interactions between sex/menopause status and fibrosis. Of these, three loci were significant in postmenopausal women, two in both men and postmenopausal women, and one exclusively in premenopausal women. These findings highlight the importance of considering sex and menopause status in genetic studies on MASLD progression.

## 6. MASLD in Male and Female Hypogonadism and Effects on MASLD of Sex Hormone Replacement Therapy and Contraceptive Use

### 6.1. Testosterone Replacement Therapy in Men

Testosterone replacement therapy (TRT) for men with hypogonadism can ameliorate sexual function, mood, energy, bone density, and muscle mass, but it may also lead to erythrocytosis, prostate enlargement, testicular shrinkage, reduced fertility, and method-specific side effects [131]. Therefore, TRT should only be considered for symptomatic men with testosterone deficiency after evaluating individual risks and benefits. Data on the hepatic effects of TRT in hypogonadal men with MASLD are limited. A 40-week randomized, double-blind, placebo-controlled trial analyzed testosterone undecanoate via the intramuscular route in hypogonadal men with T2D and over 5% hepatic steatosis at baseline. Thirty-nine subjects underwent liver MRIs [132]. By week 40, TRT recipients showed a median absolute liver fat reduction of 3.5% compared to a 1.2% increase in the placebo group (a difference of 4.7%, *p* < 0.001). Adjusted for the level of liver fat at baseline, TRT was associated with a relative reduction of 38.3% (95% CI: 25.4–49.0%, *p* < 0.001) [132]. This preliminary study suggests that TRT reduces hepatic steatosis in men with diabetes and hypotestosteronemia. Recent data suggest that the potentially beneficial effects of TRT may extend beyond MASLD to individuals with cirrhosis, where low testosterone levels result from hypogonadotropic hypogonadism combined with increased peripheral conversion of androgens to estrogen [133]. Given the limitations of previous studies in the field, Tapper et al. [134] conducted an emulated clinical trial of 282 individuals with cirrhosis and hypogonadism. They found that TRT was linked to lower mortality and reduced risk of decompensation events, especially ascites requiring paracentesis and variceal hemorrhage, without increasing the risk of HCC. The authors recommend routine endocrine screening in hepatology and considering TRT for eligible patients, particularly older adults without alcohol use. Further randomized trials are needed to determine the optimal TRT dosing, administration, and its role in alcohol-related liver disease [134,135].

### 6.2. Efficacy and Safety of Testosterone Replacement Therapy in Women

Testosterone plays a significant role in female sexual anatomy, physiology, and behavior, with levels decreasing as women age due to the loss of ovarian function, such as what occurs during menopause [136]. TRT is not recommended for women solely to address low testosterone levels, as systemic levels do not consistently correlate with hypoactive sexual desire disorder. Instead, concentrations of hormones in the central nervous system are more relevant. TRT should only be considered after other competing causes of low libido have been ruled out through a biopsychosocial approach. It is advisable to first try conventional HRT, as recommended by the NICE Menopause Guideline and the British Menopause Society [137]. Randomized controlled trials have not shown that TRT provides benefits for cognition, mood, energy, or musculoskeletal health in women [137].

### 6.3. Estradiol Replacement Therapy in Women

#### 6.3.1. Estradiol Replacement Therapy in Menopause

Little is known about the effects of estradiol replacement therapy (ERT) on MASLD and MASH, and the route of administration may be relevant. In a study by Kim et al. [138], 368 postmenopausal women receiving menopausal hormone therapy (MHT) were examined. They were divided into transdermal (n = 75) and oral (n = 293) estrogen groups. After 12 months, MASLD prevalence decreased from 24% to 17.3% in the transdermal group with no major changes in laboratory results. In contrast, MASLD prevalence increased from 25.3% to 29.4% in the oral group, which showed higher levels of triglycerides and HDL cholesterol. Interestingly, the dose and type of progestogen did not have a significant impact on the oral group. The findings of this study are compatible with the notion that transdermal estrogen may have a beneficial effect on MASLD progression.

#### 6.3.2. Estradiol Replacement Therapy in Transgender Women

Transgender women with HIV undergoing retroviral therapy and gender affirming hormone therapy (GAHT) present a complex group for analysis, but they could offer valuable insights into the effects of exogenous sex hormones on MASLD. A study of 194 participants [139] found that transgender women on GAHT had less liver steatosis compared to those not on GAHT, although MASLD severity remained higher than in cisgender men and women. These findings highlight the need for further research to understand how estrogen administration and androgen deficiency impact liver health in transgender women. Additionally, a case study reported the resolution of MASH after GAHT in a 16-year-old transgender female patient with laboratory and radiographic evidence of MASH [140]. More studies are necessary to determine if this approach is applicable more broadly, especially considering that ERT may have varying effects in pre-menopausal individuals compared to postmenopausal individuals.

#### 6.3.3. Hormonal Replacement Therapy in Turner Syndrome

Turner Syndrome (TS) is a disorder of sex chromosomes that affects 50 per 100,000 females who have only one intact X chromosome, while the second sex chromosome is missing (completely or partially. TS subjects may exhibit short stature, infertility, and cardiac complications [141]. Women with TS also exhibit various liver abnormalities, ranging from mild asymptomatic hypertransaminasemia to more severe conditions such as steatotic liver disease, hepatitis, cirrhosis/fibrosis, and liver tumors/malignancies [142,143]. Studies have shown that SLD and fibrosis are significantly more common in individuals with TS compared to controls [144].

In addition to improving laboratory abnormalities and slowing liver changes, ERT has been shown to lower overall and cardiovascular mortality rates in TS patients. Hormone replacement therapy may help reduce these risks by providing additional health benefits [142]. A study involving 82 participants with TS (71% of whom were receiving ERT) and 59 female controls demonstrated that subjects with TS had higher hepatic enzyme levels and an increased risk of hepatic dysfunction compared to the control group [145]. These changes are believed to be linked to chronic low-grade inflammation, as inflammatory biomarkers like CRP and sCD163 have shown strong correlations with liver parameters in individuals with TS [145].

### 6.4. Hepatic Effects of Progesterone Treatment

Progesterone, a vital reproductive hormone and precursor to other steroids, regulates transcription by binding to its receptors. Despite its importance, its function in the liver remains relatively unexplored [63]. Progesterone treatment can cause a temporary increase in liver enzymes, which usually resolves with dose adjustment or discontinuation of the medication. Severe liver damage is rare and is often linked to estrogenic metabolites. The risk of liver injury increases with high doses, prolonged use, or when combined with high-dose estrogens or tamoxifen [146]. Progesterone treatment can exacerbate steatosis by promoting intrahepatic DNL, contributing to MASLD pathogenesis [63].

### 6.5. Safety of Sex Hormone Replacement Therapy and Contraceptives Among Those with Chronic Liver Disease

According to the AASLD practice guidelines, women with multiple cardiovascular risk factors should avoid combined hormonal contraception (CHC). Therefore, accurate monitoring of metabolic profiles is necessary for women with MASLD [133]. MASLD itself does not contraindicate CHC, but its efficacy and safety need to be carefully considered in this patient population, which often has a significant burden of metabolic comorbidities [147]. Progestin-only contraceptives, which eliminate estrogen-related risks, include the progestin-only pill (known as the “minipill”), depot medroxyprogesterone acetate injection, subcutaneous implant, and hormonal intrauterine devices. The subcutaneous implant and levonorgestrel intra-uterine devices have the lowest failure rates and are safe regardless of the type and severity of underlying liver pathologies [147].

### 6.6. Lessons from Primary Biliary Cholangitis (PBC) and Autoimmune Hepatitis (AIH)

PBC and AIH are two rare autoimmune liver conditions that exhibit strongly sex-biased epidemiological features [148]. PBC typically shows a female-to-male (F/M) ratio of 9:1 observed in several studies, although presently the F/M ratio tends to be lower than previously reported [148]. Both ERα and ERβ modulate the response of cholangiocytes to damage by activating intracellular cascades involving extracellular regulated kinase 1/2 (ERK1/2) and phosphatidylinositol-3′kinase (PI3-kinase/AKT), which are involved in the signaling pathways of growth factors and endothelial factors implicated in the proliferation of cholangiocytes [148]. Furthermore, high estrogen levels drive the progression of PBC via the transition of the immune response from the Th1 to the Th2 phenotype [148]. Moreover, the finding that patients with severe grade PBC lose the expression of the ER indirectly indicates that estrogens per se regulate the homeostasis of cholangiocytes [148]. A growing body of evidence also suggests that testosterone may play a role in PBC pathobiology, with low testosterone levels correlating to higher levels of pro-inflammatory cytokines in female patients, while high testosterone levels play an anti-inflammatory action on T cells [148].

In AIH, observations of patient health conducted during pregnancy provide insight into the role of sex hormones. AIH severity typically diminishes during pregnancy, with disease exacerbations and flares being observed in up to half of cases after delivery [148]. Mechanistically, by acting on dendritic cells, estrogens affect the action of the immune system activity, culminating in increased functionality of T helper 1 (Th1) and up-regulation of pro-inflammatory interleukins [149]. In turn, androgens may affect the pathobiology of AIH by shifting the Th-1 to the Th-2 phenotype, thus diminishing the number of intrahepatic Th-17 and increasing the secretion of IL-10 [148].

Further evidence for the role played by sex hormones in PBC and AIH comes from an improved disease course and profound changes in T cell states in a trans man with AIH/primary sclerosing cholangitis (PSC) variant syndrome receiving GAHT [150].

## 7. Conclusions and Outlook

Over the past few years, biomedical investigation has generated a significant amount of data focusing on the development of personalized, stratified, and precision medicine [151]. Despite extensive research into patient-oriented approaches, key differences in metabolic and pathological processes between males and females are only partially defined [152]. Sexual dimorphism in MASLD explains many disease features, including disease development, progression, comorbidities, outcomes, and response to lifestyle changes and pharmacotherapies [153,154,155,156]. This sexual dimorphism is influenced by sex chromosomes and sex hormones in the context of reproductive status. Contrary to these increasingly recognized notions, none of the current guidelines from EASL, KASL, and the Latin-American working group specifically address sex-specific approaches in this field [157,158,159].

In this review, we have focused on the role of sex hormones and reproductive status to pinpoint the molecular biology of sex hormones and their receptors. We have presented the physiological grounds supporting the theory that the liver is intrinsically sexually dimorphic. Additionally, we have analyzed and commented on the abundant observational studies describing estrogen, testosterone, and SHBG in MASLD. We have also discussed the less numerous Mendelian Randomization studies. Furthermore, we have highlighted the effects of sex hormones on MASLD in pregnancy, lactation, and menopause, as well as hypogonadism in both sexes. We have also discussed the effects of sex HRT, GAHRT, and contraceptive use on MASLD. Finally, we have analyzed the hepatic effects of Progesterone treatment and the safety of sex HRT and hormonal contraceptives among those with MASLD. The data lead us to conclude that hepatic health is closely linked to sex hormones in both sexes and that hormonal manipulation might potentially reveal innovative avenues for treating MASLD in men and women. A bullet-point roadmap that may guide future research includes (i) the necessity of sex-stratified clinical trials, (ii) long-term safety assessments of HRT, and (iii) integration of sex as a biological variable in basic and clinical studies.

## Figures and Tables

**Figure 1 ijms-26-09594-f001:**
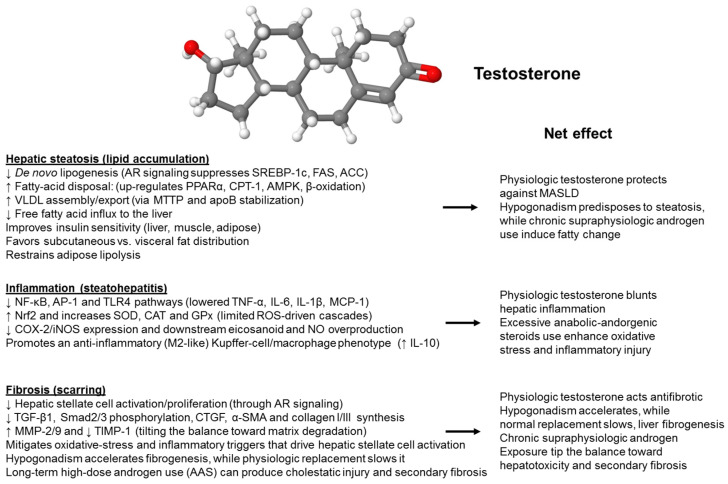
Effects of testosterone on the liver. Testosterone modulates all stages of metabolic dysfunction-associated steatohepatitis in a dose-dependent manner. The graph summarizes how natural androgen signaling protects the liver, whereas excessive use of anabolic-androgenic steroids can negate these benefits. The anti-steatotic pathway involves the activation of the androgen receptor (AR) by testosterone, which suppresses fat production (↓ SREBP-1c, FAS, ACC), promotes fatty-acid oxidation (↑ PPARα, CPT-1, AMPK), and enhances VLDL export (↑ MTTP, apoB), ultimately reducing triglyceride accumulation in the liver. In the anti-inflammatory pathway, testosterone decreases oxidative stress by activating Nrf2 to produce SOD, CAT, and GPx, inhibits NF-κB/TLR4 signaling, reduces recruitment of Kupffer cells and neutrophils, and lowers levels of pro-inflammatory cytokines (TNF-α, IL-6, IL-1β), thus preventing the progression to steatohepatitis. The anti-fibrotic pathway involves AR signaling in hepatic stellate cells, inhibiting activation and growth (↓ TGF-β1, p-Smad2/3, α-SMA, collagen I/III) while promoting matrix breakdown (↑ MMP-2/9, ↓ TIMP-1), which slows fibrosis development. Physiological testosterone levels protect the liver, whereas testosterone deficiency accelerates steatosis, inflammation, and fibrosis progression. Chronic high-dose anabolic-androgenic steroid (AAS) use can worsen these conditions. Abbreviations used: α-SMA, α-smooth muscle actin; ACC, acetyl-CoA carboxylase; AMPK, AMP-activated protein kinase; apoB, apolipoprotein B; CAT, catalase; CPT-1; carnitine palmitoyltransferase I; FAS, fatty acid synthetase; GPx, glutathione peroxidase; IL-1β, interleukin-1β; IL-6, interleukin 6; MMP, matrix metalloproteinase; MTTP, microsomal triglyceride transfer protein; NF-κB, nuclear factor ‘kappa-light-chain-enhancer’ of activated B-cells; Nrf2, Nuclear factor erythroid 2-related factor 2; p-Smad2/3, phosphorylated Smad 2 and Smad3; PPARα, Peroxisome proliferator-activated receptor-α; SOD, superoxide dismutase; SREBP-1c, sterol regulatory element-binding protein-1c; TGF-β1, transforming growth factor-β1; TIMP-1, tissue inhibitor of metalloproteinases-1; TLR4, toll-like receptor 4; TNF-α, tumor necrosis factor-α; VLDL, very low density lipoprotein.

**Figure 2 ijms-26-09594-f002:**
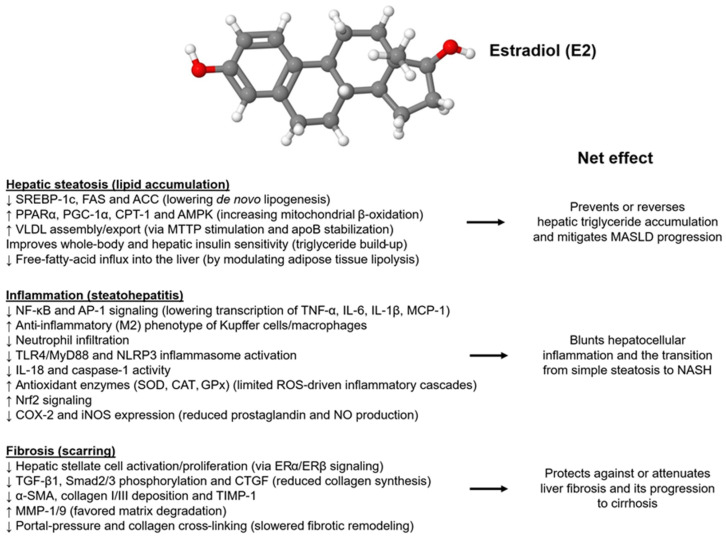
Estrogen (17β-estradiol, E2) provides comprehensive protection for the liver. This diagram illustrates the main ways in which estradiol (ER) counteracts the three stages of fatty liver disease. Anti-steatotic effects: ERα/ERβ signaling suppresses the lipogenic program (↓ SREBP-1c, FAS, ACC), enhances fatty acid oxidation and mitochondrial biogenesis (↑ PPARα, PGC-1α, CPT-1, AMPK), and promotes VLDL export (↑ MTTP, apoB), collectively reducing triglyceride accumulation. Anti-inflammatory effects: Estradiol reduces oxidative stress (Nrf2-driven induction of SOD, CAT, GPx), inhibits NF-κB, AP-1, TLR4, and inflammasome pathways, decreases Kupffer cell/neutrophil recruitment, and lowers pro-inflammatory cytokines (TNF-α, IL-6, IL-1β), preventing progression from simple steatosis to MASH. Anti-fibrotic effects: Direct ER-mediated inhibition of hepatic stellate cell activation (↓ α-SMA, TGF-β1, p-Smad2/3, CTGF), re-balancing extracellular-matrix turnover (↑ MMP-1/9, ↓ TIMP-1), and limiting collagen I/III deposition. Overall, estradiol slows or reverses hepatic steatosis, inflammation, and fibrosis, providing a basis for sex differences and estrogen-based treatments in MASLD/MASH. Abbreviations used: α-SMA, α-smooth muscle actin; ERα/β, estrogen receptor α/β; ACC, acetyl-CoA carboxylase; AMPK, AMP-activated protein kinase; apoB, apolipoprotein B; CAT, catalase; CPT-1; carnitine palmitoyltransferase I; CTGF, cellular communication network 2; FAS, fatty acid synthetase; GPx, glutathione peroxidase; IL-1β, interleukin-1β; IL-6, interleukin 6; MMP, matrix metalloproteinase; MTTP, microsomal triglyceride transfer protein; NF-κB, nuclear factor ‘kappa-light-chain-enhancer’ of activated B-cells; Nrf2, Nuclear factor erythroid 2-related factor 2; p-Smad2/3, phosphorylated Smad 2 and Smad3; PPARα, Peroxisome proliferator-activated receptor-α; SOD, superoxide dismutase; SREBP-1c, sterol regulatory element-binding protein-1c; TGF-β1, transforming growth factor-β1; TIMP-1, tissue inhibitor of metalloproteinases-1; TLR4, toll-like receptor 4; TNF-α, tumor necrosis factor-α.

**Figure 3 ijms-26-09594-f003:**
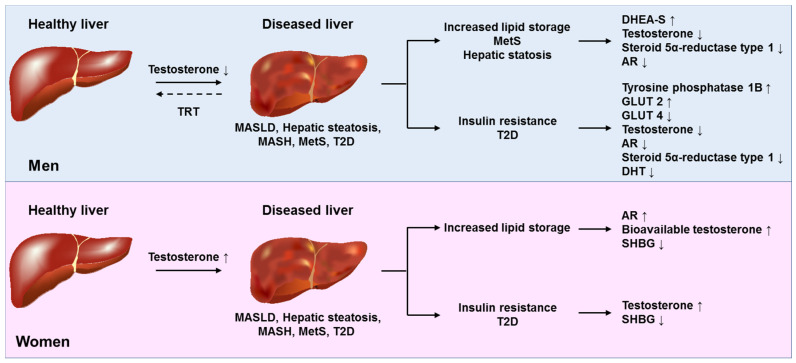
Molecular effects of testosterone on hepatic lipid metabolism. In males, low circulating testosterone levels are associated with hepatic pathology characterized by increased lipid storage, metabolic syndrome (MetS), hepatic steatosis, impaired insulin signaling, and subsequent type 2 diabetes (T2D) with numerous molecular changes. Testosterone replacement therapy (TRT) can partially restore these changes. In females, elevated (supraphysiological) testosterone levels are linked to liver pathology through enhanced lipid accumulation, disturbed glucose metabolism, and a higher risk of T2D. Abbreviations: AR, androgen receptor; DHEA-S, dehydroepiandrosterone sulfate; GLUT2/4, glucose transporters 2 and 4; MASH, metabolic dysfunction-associated steatohepatitis; MASLD, metabolic dysfunction-associated steatotic liver disease; MetS, metabolic syndrome; SHBG, sex hormone-binding globulin. This figure was adapted and simplified from a review article by Kasarinaite et al., providing additional insights on how sex hormones influence liver metabolism, immune responses, and the development of conditions such as MASLD, MASH, and fibrosis, with an additional focus on experimental models and cell-based modeling of MASLD [62].

**Figure 4 ijms-26-09594-f004:**
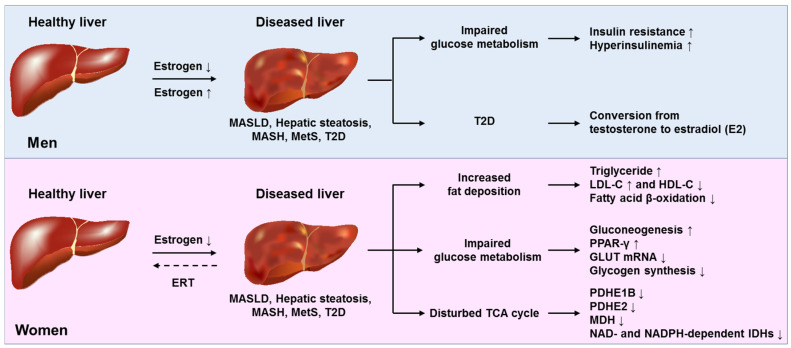
Molecular effects of estradiol on hepatic lipid metabolism. In males, impaired estrogen signaling predisposes to liver disease. Low estrogen worsens hepatic dysfunction by affecting glucose metabolism, while supraphysiological estrogen levels can contribute to MASLD in individuals with type 2 diabetes (T2D). In females, low levels of circulating estrogen are associated with liver issues, including increased fat storage and disruptions in glucose and tricarboxylic acid (TCA) cycle pathways. Estrogen replacement therapy (ERT) can help restore some of these changes. Abbreviations: ERT, IDH(s), isocitrate dehydrogenase(s); MASH, metabolic dysfunction-associated steatohepatitis; MASLD, metabolic dysfunction-associated steatotic liver disease; MDH, Malate dehydrogenase; MetS, metabolic syndrome; LDL-C, low-density lipoprotein cholesterol; HDL-C, high-density lipoprotein cholesterol; PPAR-γ, peroxisome proliferator-activated receptor-γ; GLUT, glucose transporter; PDHE1B, pyruvate dehydrogenase E1 beta; PDHE2, pyruvate dehydrogenase complex subunit E2. This figure was adapted and simplified from [62].

**Table 1 ijms-26-09594-t001:** Epidemiological, observational, and meta-analytic evidence supporting the role of serum concentrations of endogenous sex hormones and sex hormone-binding globulin in MASLD.

Author, Year [Ref]	MASLD/MASH Diagnosis	Origin	Patient Population	Findings	Conclusions
Yang JD, 2014 [92]	LB	USA	541 adults with MASH.	ACOR and 95% CI for more severe fibrosis were 1.4 (0.9, 2.1) (*p* = 0.17) for postmenopausal women and 1.6 (1.0, 2.5) (*p* = 0.03) for men, with premenopausal women as the reference. The ACOR and 95% CI of having more severe fibrosis in men than women were 1.8 (1.1, 2.9) for patients below 50 years (*p* = 0.02) and 1.2 (0.7, 2.1) for patients over 50 years (*p* = 0.59).	Men have higher odds of more severe fibrosis than pre-menopausal women; post-menopausal women have liver fibrosis of severity like men.
Klair JS, 2016 [93]	LB	USA	488 women in post-menopause with MASLD.	After adjusting for multiple confounding factors, premature menopause was a risk factor for more severe fibrosis. Time from menopause was associated with more severe fibrosis.	In MASLD women, the duration of postmenopausal estrogen deficiency increases the odds of liver fibrosis.
Sarkar M, 2017 [94]	Non-contrast abdominal CT scan with liver attenuation ≤ 40 HU after excluding competing causes of SLD.	USA	1052 women participating in the prospective population-based multicenter CARDIA study, whether cFT levels measured at year 2 were associated with prevalent MASLD at year 25.	Increasing quintiles of cFT were associated with the prevalence of MASLD at Year 25, regardless of confounders. This association was confirmed among 955 women who did not have any androgen excess and was partially mediated by VAT volume.	Increasing cFT is associated with the prevalence of MASLD in middle-aged women, even in the absence of androgen excess, mediated by visceral adiposity.
Yang JD, 2017 [95]	LB	USA	1112 patients with MASLD participating in 3 large U.S. studies.	Premenopausal women, compared to men, had a higher risk of LOBI, hepatocyte ballooning, and Mallory-Denk bodies. Compared to postmenopausal women, they also had an increased risk of LOBI and Mallory-Denk bodies. In premenopausal women, oral contraceptives were associated with an increased risk of LOBI and Mallory-Denk bodies. In postmenopausal women, HRT was associated with an increased risk of LOBI.	Being a premenopausal woman or a female user of synthetic hormones is associated with increased histologic severity of hepatocyte injury and inflammation among patients with MASLD.
Minato S, 2018 [96]	Surrogate biomarkers	Japan	Retrospective analysis of 102 reproductive-aged women with a confirmed diagnosis of PCOS (ICD-10 codes).	Raised liver enzymes were found in 33.3% of cases. PCOS subjects had significantly higher BMI values than those with normal liver enzymes. In ROC analyses, T proved to be related to SLD.	An algorithm using BMI, glycemia, and testosterone levels may predict raised liver enzymes in PCOS women.
Mueller NT, 2020 [97]	LB	USA	573 children and adolescents aged 18 or younger.	In both sexes, lower SHBG was inversely associated with steatosis severity and with portal inflammation in girls. Higher T was associated with improved steatosis and fibrosis in boys but was detrimental in girls. In both sexes, higher estrone, estradiol, and T were associated with a lower grade of portal inflammation; higher estradiol was positively associated with the severity of ballooning; DHEAS was inversely associated with ballooning and MASH severity.	Sex hormones are associated with MASLD histological features in children and adolescents.
Sarkar M, 2021 [98]	LB	USA	159 random men, participating in the MASH CRN database.	Low cFT was associated with MASH, independent of age, WC, insulin resistance, and TG, and higher liver fibrosis stages	In men, low cFT is independently associated with MASH presence and severity.
Wamg J, 2021 [99]	Cases were identified using Medicare claims data; controls were selected among participants without liver disease.	USA	Nested case–control study with 1861 cases and 17,664 controls in the Multiethnic Cohort Study.	There was an inverse relationship between later age at menarche and MASLD (Ptrend = 0.01). Parity was associated with an increased risk of NAFLD. The use of oral contraceptives was associated with a higher odds of MASLD. Duration of use, women with oophorectomy or hysterectomy had a higher MASLD risk than women with natural menopause. A longer duration of menopause hormone therapy (only estrogen therapy) was associated with an increasing risk of MASLD.	Menstrual and reproductive factors, along with exogenous hormones, are associated with the risk of MASLD.
Dilimulati D, 2021 [100]	FibroScan	China	360 adults with obesity were enrolled, with follow-up data available for 132 individuals who underwent LSG.	In the preoperative cohort, lower TT was associated with higher CAP and LSM in men. In women, higher TT was associated with higher CAP. In the postoperative cohort, changes in TT levels at 3 months after surgery were negatively correlated with changes in CAP values in men, and in women, changes in TT levels at 6 months post-surgery were positively correlated with variations in CAP values. After adjustment for confounding factors, the variations in TT levels were independently associated with CAP variations in both sexes.	TT concentrations are involved in pre-operative MASLD and post-operative disease regression in both sexes.
Mo MQ, 2022 [101]	USG in most studies and LB in three studies.	China	Meta-analysis of 2995 MASLD patients from 10 published cross-sectional studies.	Among men with MASLD, those with moderate-severe disease had lower TT than those with mild liver disease. TT and SHBG were significantly associated with moderate-severe MASLD. Among men older than 50, SHBG levels were lower in those with moderate-severe disease; among men with BMI > 27 kg/m^2^, moderate-severe MASLD was associated with higher SHBG levels than those with mild disease.	In men, lower TT is associated with more severe MASLD. However, the relationship of SHBG with MASLD severity or MASLD remains uncertain.
Cao W, 2022 [102]	USG	China	732 participants aged 50–80 years were enrolled from communities.	After adjusting for confounders, LRA found a negative correlation of SHBG with MAFLD in men. Among women, SHBG and FSH had a negative correlation with MAFLD. In multivariate linear regression analysis, SHBG was a negative factor for LFC in both sexes. In women, FSH was a borderline significant negative factor for LFC. SHBG was negatively correlated with MAFLD in middle-aged and elderly individuals of both sexes. In women, FSH was negatively correlated, and bioactive testosterone was positively correlated with MAFLD.	Sex hormones are associated with MAFLD.
Zhang X, 2022 [103]	USG	China	1155 subjects with T2D.	In men with T2D, increasing TT values were associated with decreased odds of MASLD. There were no statistically significant correlations observed between rising concentrations of androgen precursors and the likelihood of MASLD (all *p* values > 0.05). Among women with T2D, no significant associations were found between TT, androstenedione, DHEA, and DHEAS, with the risk of MASLD.	Serum TT is strongly associated with MASLD in men with T2D.
Zhang Z, 2022 [104]	Probable MASH was defined by concurrent NAFLD and MetS.	China	Cross-sectional study enrolling 1782 men with T2D.	TT quartiles were associated in a negative manner with probable MASH and disease inflammatory progression, but positively with fibrotic progression. In stratified analyses, the interactions of age, duration of T2D, and dyslipidemia were significant for inflammatory progression rather than for fibrotic progression.	In men, TT exhibits variable relationships with inflammatory and fibrotic components in MASLD, implying that this hormone has different roles in the individual features of MASLD histology.
Yang LJ, 103 [105]	USG	China	Cross-sectional study involving 1005 men with T2D.	After adjustments for confounders, the top TT tertile, compared to the lowest tertile, was associated with a reduced prevalence of MASLD. TT and MASLD were more strongly associated in lean individuals than among subjects with overweight/obesity. A significant interaction of TT with overweight/obesity (*p* for interaction = 0.018 for MASLD) was found.	Higher serum concentrations of TT were associated with a significantly lower prevalence of MASLD among men with T2D. The association of TT with MASLD was stronger in lean subjects.
De Herdt C, 2023 [106]	LB	Europe	Retrospective analysis of 134 men who underwent metabolic-hepatological work-up and liver biopsy.	No significant differences were found in concentrations of TT and cFT between MASL and MASH, and steatosis and ballooning. cFT was significantly lower in a higher stage of fibrosis (*p* = 0.013), not TT, and this difference did not persist after controlling for metabolic onfounders. A higher stage of LOBI was associated with lower TT concentrations (*p* = 0.033), not cFT, and this difference did not persist after controlling for VAT surface and HOMA-IR.	No association has been found between testosterone levels and MASLD, histological subgroups, or fibrosis. The lower levels of cFT observed among subjects with higher liver fibrosis stages and the association of TT with LOBI are driven by metabolic dysfunction.
Apostolov R, 2023 [107]	Cirrhosis was confirmed by a hepatologist through a combination of clinical, biochemical, radiological, and pathological findings.	Australia	Monocentric retrospective survey of 766 men with cirrhosis, with ALD and MASLD accounting for 33.3% and 11.9% of cases, respectively, in whom the determination of TT levels was available.	Low TT levels and cFT levels were found in 53.3% and 79.6% of cases, respectively. Median TT was lower in men with ALD and MASLD than in cirrhosis owing to other etiologies, irrespective of age and MELD score. TT was associated in an inverse manner with 1-year mortality or transplant and liver decompensation.	Hypotestosteronemia is common among men with cirrhosis and is associated with unfavourable clinical outcomes. Subjects with ALD and MASLD exhibit significantly lower TT serum concentrations compared to other causes of cirrhosis.
Cai X, 2023 [108]	FLI	Europe	Observational study involving 2239 participants followed up for an average of 6.5 years.	In this observational study, in men, TT, DHT, progesterone, and 17-OHP were inversely associated with FLI. Among women, free T was positively associated with FLI. SHBG was inversely associated with FLI across sexes. At MR analysis, no causal association was identified between genetically determined sex hormones and LFC. However, higher genetically determined SHBG was related to lower LFC in women.	In women, SHBG helps protect against liver fat accumulation.
Maldonado SS, 2024 [109]	LB	USA	205 MASLD participants in the CRN.	After adjustment for confounders, higher AMH quartiles were inversely associated with MASLD histological features, including prevalent MASH, NAS ≥ 5, Mallory hyaline, and higher fibrosis stage.	Aging of the reproductive system is associated with the histologic severity of MASLD in women.
Weng C, 2024 [110]	ICD-9 (571.8) and ICD-10 (K76.0, K75.8) from the hospital admissions and death records	China	187,395 men and 170,193 women from the UK Biobank followed up for 12.49 years using linear and nonlinear Cox regression models and MR analysis to test associations.	During follow-up, 2209 men and 1886 women with MASLD were identified. Elevated SHBG levels were linearly associated with a reduced risk of MASLD in women, but not in men. Higher BAT levels were associated with a reduced risk of MASLD in men and an increased disease in women. Genetically determined SHBG and BAT levels were linearly associated with MASLD risk in women; in men, an “L-shaped” MR association between SHBG levels and MASLD risk was found. Bidirectional MR analysis confirmed that MASLD was causally associated with SHBG and BAT levels in either sex.	In women, lower SHBG and higher BAT levels confer an increased risk of MASLD, both at conventional analysis and with MR assessment. In men, SHBG acts in a nonlinear manner. MASLD affects SHBG and BAT levels.
Wang Y, 2025 [111]	USG	China	155 male children with obesity, with a mean age of 11.07 ± 1.53 years.	Children with MAFLD had statistically higher BMI, fasting insulin, HOMA-IR, fasting C-peptide, WBC, HbA1c, ALT, and AST, and lower levels of HDL, T, and SHBG than controls with simple obesity. At LR BMI, testosterone, and SHBG independently predicted MAFLD in boys, and these variables are of potential value in the early diagnosis of MAFLD as indicated by ROC curve analysis.	Among boys with obesity, BMI, testosterone, and SHBG independently predict MAFLD.

Abbreviations: ACOR; adjusted cumulative odds ratio; ALD; alcohol-related liver disease; ALT; alanine transaminase; AMH, Anti-Müllerian hormone; AST; aspartate transaminase; aOR; adjusted odds ratio; BAT; bioavailable testosterone; BMI; body mass index; CAP; controlled attenuation parameter; CARDIA; Coronary Artery Risk Development in Young Adults; cFT; calculated free testosterone; CI, confidence interval; CRN; Clinical Research Network; CT; computed tomography; DHEA, dehydroepiandrosterone; DHEAS, dehydroepiandrosterone sulfate; DHT, dihydrotestosterone; FLI, fatty liver index; FSH; follicle-stimulating hormone; GWAS, genome-wide association study; HBA1c, glycated hemoglobin; HDL, high density lipoprotein; HOMA-IR, homeostasis model assessment of insulin resistance; HU, Hounsfield units; ICD-9, International Classification of Diseases; 9th Revision; ICD-10, International Classification of Diseases, 10th Revision; IQR, interquartile range; LB, liver biopsy; LFC, liver fat content; LOBI, lobular inflammation; LR, logistic regression; LRA, logistic regression analysis; LSG, laparoscopic sleeve gastrectomy, LSM, liver stiffness measurement; MAFLD, metabolic dysfunction-associated fatty liver disease; MASLD, metabolic dysfunction-associated steatotic liver disease; MELD, model for end-stage liver disease; MetS, metabolic syndrome; MR, Mendelian randomization; MRI, magnetic resonance imaging; NAS, NASFLD activity score; OR, odds ratio; PCOS, polycystic ovary syndrome; ROC, receiver operator curve; 17-OHP, 17α-hydroxyprogesterone; SHBG, sex hormone-binding globulin; SLD, steatotic liver disease; T, testosterone; T2D, type 2 diabetes; TT, total testosterone; USG, ultrasonography; VAT, visceral adipose tissue; WBC, white blood cells.

## Data Availability

No new data was generated in this study.

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
