# Peer review of "Sex Hormones and Metabolic Dysfunction-Associated Steatotic Liver Disease"

_ijms, 2025, doi:10.3390/ijms26199594_

Round 1

Reviewer 1 Report

Comments and Suggestions for Authors

This review article provides a comprehensive and elegant overview of the molecular actions and clinical significance of sex hormones in MASLD. It is a highly valuable work for readers, covering a wide spectrum from basic and epidemiological research to hormone replacement therapy. The manuscript offers important insights for both clinicians and researchers, and I believe it merits publication. However, a few revisions would further enhance its quality:

  • Regarding the association between sex hormones and MASLD progression, most of the available evidence is based on cross-sectional or retrospective analyses. It would be beneficial to emphasize in the text that longitudinal and prospective studies are needed to clarify causal relationships.
  • Please clarify whether hormone replacement therapy and sex-specific approaches to MASLD are addressed in major international guidelines (i.e., whether such recommendations exist or are currently absent). This will help readers better appreciate the current level of evidence.
  • The manuscript mentions important research needs, such as the lack of sex-specific clinical trials and the need to assess the safety of hormone replacement therapy. Summarizing these points at the end of the manuscript would provide a clearer roadmap for future studies.
  • While the review appropriately focuses on MASLD, other chronic liver diseases with marked female predominance—such as PBC and AIH—are also strongly influenced by sex hormones and sex differences. Briefly addressing these conditions would broaden the scope and clinical relevance of the review.

Author Response

Reviewer 1

This review article provides a comprehensive and elegant overview of the molecular actions and clinical significance of sex hormones in MASLD. It is a highly valuable work for readers, covering a wide spectrum from basic and epidemiological research to hormone replacement therapy. The manuscript offers important insights for both clinicians and researchers, and I believe it merits publication.

We are grateful for the reviewer’s positive assessment of our work and for the valuable suggestions that have helped us improve the clarity and breadth of the review 

However, a few revisions would further enhance its quality:

  • Regarding the association between sex hormones and MASLD progression, most of the available evidence is based on cross-sectional or retrospective analyses. It would be beneficial to emphasize in the text that longitudinal and prospective studies are needed to clarify causal relationships.

Response: We have added a dedicated sentence at lines 475-476 emphasizing the lack of longitudinal data and explicitly calling for large-scale prospective cohort studies and randomized trials to establish causality.

  • Please clarify whether hormone replacement therapy and sex-specific approaches to MASLD are addressed in major international guidelines (i.e., whether such recommendations exist or are currently absent). This will help readers better appreciate the current level of evidence.

Response: A new paragraph has been inserted in Section 7, briefly summarizing current positions of EASL, KASL, and Latin-American working group guidelines. It highlights that none of them currently issue formal recommendations on HRT in MASLD, thereby stressing the evidence gap (lines 788-790).

  • The manuscript mentions important research needs, such as the lack of sex-specific clinical trials and the need to assess the safety of hormone replacement therapy. Summarizing these points at the end of the manuscript would provide a clearer roadmap for future studies.

Response: We have revised the “Conclusions & Outlook” section to offer a succinct bullet-point roadmap that includes (i) the importance of sex-stratified clinical trials, (ii) long-term safety evaluations of HRT, and (iii) incorporation of sex as a biological variable in fundamental research (lines 804-807).

  • While the review appropriately focuses on MASLD, other chronic liver diseases with marked female predominance—such as PBC and AIH—are also strongly influenced by sex hormones and sex differences. Briefly addressing these conditions would broaden the scope and clinical relevance of the review.

Response: A brief subsection (Section 6.6) now addresses the impact of sex hormones on PBC and AIH, covering epidemiology, immune-modulatory mechanisms and therapeutic considerations (lines 753-779). This expands the clinical relevance of the review.

We believe that these revisions fully address the reviewers’ comments and substantially enhance the manuscript. We thank the reviewers once again for their constructive feedback and hope that the revised version will be deemed suitable for publication in the International Journal of Molecular Sciences.

Sincerely,

Ralf Weiskirchen

Reviewer 2 Report

Comments and Suggestions for Authors

Weiskirchen et al.'s review is interesting and very well documented. While the importance of sex and sex hormones in metabolic health and diseases has been widely explored, sex-dependent therapeutic approaches are still very scarce. I would suggest a few modifications/additions to the current review to help readers navigate the complicated relationship between sex hormones and liver diseases.

  1. The review does not mention transgender hormonal therapy, and a paragraph should be added to address this.
  2. Figures showing the effects of estradiol/testosterone on the liver would help readers to understand the key message, as in the review from Kasarinaite et al. DOI: 10.3390/cells12121604
  3. Figure 1 shows estradiol (the same as Figure 2) and discusses testosterone. Please correct this.

Author Response

Reviewer 2

Weiskirchen et al.'s review is interesting and very well documented. While the importance of sex and sex hormones in metabolic health and diseases has been widely explored, sex-dependent therapeutic approaches are still very scarce. I would suggest a few modifications/additions to the current review to help readers navigate the complicated relationship between sex hormones and liver diseases.

We appreciate the reviewer’s kind words and insightful recommendations, which have strengthened our manuscript

  1. The review does not mention transgender hormonal therapy, and a paragraph should be added to address this.

Response: We have added a new subsection (6.3.2) dedicated to transgender hormone therapy. This section summarizes the current evidence on its metabolic and hepatic effects, highlights knowledge gaps, and calls for systematic studies in this population (lines 699-711).

  1. Figures showing the effects of estradiol/testosterone on the liver would help readers to understand the key message, as in the review from Kasarinaite et al. DOI: 10.3390/cells12121604

Response: The two schematic figures (new Figures 3 and 4) have been prepared that were adapted from adapted from graphics presented by Kasarinaite et al. (Cells 2023;12:1604).

Figure 3: Molecular effects of testosterone on hepatic lipid metabolism.

Figure 4: Molecular effects of estradiol on hepatic lipid metabolism.

  1. Figure 1 shows estradiol (the same as Figure 2) and discusses testosterone. Please correct this.

Response: The figure mix-up has been resolved. Figure 1 now exclusively depicts testosterone-related pathways, while Figure 2 addresses estrogen. All captions and in-text references have been updated accordingly.

We believe that these revisions fully address the reviewers’ comments and substantially enhance the manuscript. We thank the reviewers once again for their constructive feedback and hope that the revised version will be deemed suitable for publication in the International Journal of Molecular Sciences.

Sincerely,

Ralf Weiskirchen